# Motivation, Intention and Action: Wearing Masks to Prevent the Spread of COVID-19

Geoff Kaine [1,*], Vic Wright [2] and Suz Greenhalgh [3]

1   Manaaaki Whenua Landcare Research, Hamilton 3216, New Zealand
2   UNE Business School, University of New England, Armidale 2351, Australia
3   Manaaaki Whenua Landcare Research, Auckland 1072, New Zealand
*   Correspondence: kaineg@landcareresearch.co.nz; Tel.: +64-7859-3763

**Abstract:** Governments are seeking to slow the spread of COVID-19 by implementing measures that encourage, or mandate, changes in people's behaviour such as the wearing of face masks. The success of these measures depends on the willingness of individuals to change their behaviour and their commitment and capacity to translate that intention into actions. Understanding and predicting both the willingness of individuals to change their behaviour and their enthusiasm to act on that willingness are needed to assess the likely effectiveness of these measures in slowing the spread of the virus. We analysed responses to two different regional surveys about people's intentions and behaviour with respect to preventing the spread of COVID-19 in New Zealand. While motivations and intentions were largely similar across the regions, there were surprisingly large differences across the regions regarding the frequency of wearing face masks. These regional differences were not associated with regional differences in demographics (or in Alert levels) but were associated with regional differences in the number of confirmed cases of COVID-19. The results highlight the importance to policy design of distinguishing the factors that might influence the formation of behavioural intentions from those that might influence the implementation of those intentions.

**Keywords:** mask wearing; behaviour; behavioural intentions; COVID-19; policy

## 1. Introduction

The COVID-19 pandemic is an unprecedented global crisis that triggered drastic changes in social life, personal freedoms, and economic activity. The effectiveness of measures advocated by government to slow or stop the spread of COVID-19 (coronavirus disease of 2019) depends on the commitment and capacity of individuals to comply with them, and to change their behaviour accordingly [1–3]. Poor levels of compliance can put the achievement of policy outcomes at risk [4,5]. For example, failure to wear face masks, self-isolate when unwell and socially distance may put satisfaction of a government policy objective of slowing the spread of COVID-19 at risk with the result that considerable resources must be invested in enforcement, and possibly the imposition of lockdowns causing both economic [6] and psychological damage [7] to avoid higher rates of infection and mortality.

When the advocated measures entail novel behaviour for people, novel either in nature (such as wearing face masks) or degree (such as more careful and sustained isolation when unwell), knowing the degree to which individuals are willing to modify their behaviour is especially important for predicting how effective measures such as mask wearing are likely to be. Unfortunately, intentions do not necessarily translate into action. Hence, understanding why individuals may not change their behaviour, despite their reported intentions to do so, is also crucial if policies to encourage measures like wearing face masks and social distancing are to be effective.

Building on previous research [8–11], we draw on the social psychological concepts of involvement and attitudes in this paper to predict the willingness of the public in several

regions across New Zealand to prevent the spread of COVID-19 by wearing face masks, self-isolating when unwell, and getting tested for COVID-19. We then investigate differences across the regions in the extent to which intentions to wear face masks translated into mask wearing behaviour. Our aim was to explain why, despite similarity in people's intentions regarding preventing the spread of COVID-19, there were surprisingly large differences in their behaviour with respect to wearing face masks.

## 2. Background

### 2.1. Theory

Research and analysis focused on the extended decision-making process that directs the non-routine actions of individuals has long included recognition that there are two phases to the process: decision and implementation. Extended decision-making processes are triggered when, for example, individuals experience novel situations (such as a pandemic) that require considering changing routine behaviours. The natural point of separation between these phases is the 'action intention' which arises once a decision is made. This action intention, which can be generic or specific, is normally referred to as 'behavioural intention' in the literature [12–15]. This intention is the new action or actions (e.g., booking flights, attending an aerobics class, wearing a face mask) the individual intends to undertake to meet a new, personal aspiration (e.g., have a holiday, improve health, avoid COVID-19 infection, respectively).

Behavioural intention, rather than actual behaviour, is usually the appropriate result on which to focus when seeking to analyse decision-making because related, subsequent actual behaviour can diverge from intended behaviour for a wide variety of reasons. Seeking to explain actual behaviour with reference to decision processes triggered by a decision problem could unhelpfully conflate decision considerations and implementation considerations, masking the contribution of the former. In effect, the contemporary approach is to distinguish decision making from decision implementation.

While factors that may impact upon implementation could well figure in the decision-making process (particularly as costs or benefits attaching to decision options), substantive modelling of implementation is required to complete the analysis, or projection, of actual behaviour flowing from behavioural choices made in response to perceived decision problems.

In any specific applied setting, particularly those involving existing practices and products, decision implementation is routine and familiar to all users. In the case of novel practices and products, decision implementation assumes greater importance because it defines the rate of adoption of the novel behaviour or product. Measures such as wearing face masks and social distancing that were introduced to contain the spread of COVID-19 fit into this novel category: when introduced these practices were new to each person, even the advice to wash hands much more regularly is novel, in degree, and the wearing of face masks obviously so for most people in New Zealand. Hence, when new practices are being introduced to cope with a pandemic, whose impacts are driven by rates of transmission across a population, rates of practice adoption matter a good deal (as reflected by concerns about 'vaccine hesitancy' [16,17]).

In prior studies [8–11] we investigated behavioural intentions, and actual behaviour, with respect to measures advocated by government to suppress or eliminate COVID-19 in New Zealand. The central rationale for the $I_3$ model we use (see Section 3.1 below) is that scarce cognitive attention is allocated variously to novel actions according to the felt relevance of those actions to the needs of the individual. Involvement, a concept drawn from social psychology and marketing [18], is used in the $I_3$ model as a multi-dimensional measure of felt needs (functional, experiential or self-expressive). Clearly, people's involvement with a subject will depend on their beliefs about how that subject affects the satisfaction of needs, and their evaluative judgements about those beliefs. In short, involvement is a measure of motivation [19].

In the context considered here, the essence of the $I_3$ model is that the people's behavioural intentions (to wear a face mask, for example) depend on the strength of their

motivation to engage with the policy outcome (eliminating COVID-19 from New Zealand) as well as the strength of their motivation to observe the policy measure (wear a face mask). In $I_3$ analyses people are allocated into quadrants based on these two motivations. The quadrants are then analysed to identify, within each quadrant, the diversity and roles of salient beliefs and attitudes, and whether quadrant members were favourably or unfavourably predisposed to comply with the policy measures. This approach was designed to identify policy and promotional activity that could be implemented to strengthen desired behavioural intentions, and actual compliance, with respect to the policy measures (in the context of current implementation of behavioural intentions and salient attitudes and beliefs). The research was not designed to model decision-making processes per se.

In two of the previous studies [8,9] behavioural intentions with respect to wearing face masks, self-isolating and getting tested for COVID-19 were investigated, together with self-reports of actual behaviour with respect to wearing face masks. The studies were conducted using the same questionnaire but in different regions of New Zealand. The results in these two studies indicated that, despite strong similarities across regions in behavioural intentions, there were marked dissimilarities in actual behaviour [9]. In particular, while the willingness to act to prevent the spread of COVID-19 was similar across the regions, the wearing of face masks was dramatically different across the regions [9].

Where, as in this instance, diversity in actual behaviour occurs in a context of shared behavioural intentions with respect to a novel behaviour, it is necessary to identify the cause. First, because, if the claimed integrity of the $I_3$ model [8,9,11,20–24], and other models of behavioural intentions [12,13], is to be sustained, it is necessary to be able to identify a plausible and active cause for hesitancy to implement an intention. In a basic sense, behavioural intention is not the sole appropriate policy target if it is not highly, positively correlated with actual behaviour. Second, because, for any additional actions to be taken to accelerate adoption of the behaviour to be appropriate, the causes of the hesitancy need to be identified.

Bagozzi [12], one of the few theorists to model the implementation of behavioural intentions, has suggested there is a planning and control loop that can be identified which governs the translation of intention into (new and ongoing) behaviour. This loop determines the 'how' and 'when' of implementation, and the monitoring of initial actions regarding the adequacy of goal attainment and any unanticipated outcomes. This outcome information is fed back into both the decision-making and implementing sub-systems.

Bagozzi [12] draws attention to the fact that different sets of, sometimes partly overlapping, factors influence the formation of behavioural intentions and their implementation. This has consequences for understanding what factors can properly be said to act as 'barriers' [12] to desired behaviour changes. The notion of a 'barrier' is often unhelpfully broad because, as here, the set of factors serving to create a favourable or unfavourable attitude to novel behaviour can, and in this case must, be different to the set of factors seeming to impede behaviour change. Particularly, 'barriers', in normal usage, usually refers to things or situations impeding movement in an intended direction: intended by the subject, not some observer. 'Barriers' has relevant meaning, therefore, as factors causing behavioural intention to not lead to the behaviour in question. Arguably, its use with respect to the forming of behavioural intentions reveals more about observer preferences than impediments to implementation the subject confronts.

In the case of mask wearing, the barrier to continuous action may be non-availability of masks, unanticipated social opprobrium when they are worn or unexpected discomfort (both the latter two reflecting inaccurate judgement in arriving at behavioural intention). The most obvious, and logically the first, 'barrier' to seek out is the absence of a behavioural trigger. Identifying behavioural intention by questioning subjects rather than tracking behaviour, is to discover a predisposition to act. A failure to act implies the absence of a trigger to activate the predisposition. That is, levels of involvement have been sufficient for attention to be triggered and personally appropriate responses to be identified for each salient measure to meet personal aspirations, but the need to act is not yet apparent to the individual.

In this case, if masks are readily available, socially acceptable and reasonably comfortable, the missing trigger will presumably be related to perceived need: the perceived imminent threat of airborne infection. If this threat is perceived to be high, then an intention to wear a face mask is activated and translates into action (because there is a perceived need). If the threat is perceived to be low, then an intention to wear a face mask is not activated (because there is no perceived need) and so does not translate into action.

The perceived threat of airborne infection is, inevitably, subjective and cue-driven amongst most people. The cues employed to infer the perceived threat of infection may well be influenced by reported infections in an area, trends in them and social discussion about them, and perhaps by the prevalence of mask wearing. Infection incidence may be the critical catalyst that triggers action. The adoption of behaviours such as the wearing of face masks has been associated previously with a range of variables including perceptions of the perceived risk of infection, the local incidence rate of COVID-19 and feelings of stress in relation to COVID-19 [25].

In the next section we provide a brief description of the history of COVID-19 in New Zealand to place the subsequent analysis in its proper context.

### 2.2. COVID-19 in New Zealand

The first detection of COVID-19 in New Zealand was on 28 February 2020 [26]. By the end of March, the central government had closed New Zealand's international border to all except returning citizens and permanent residents and had begun pursuing a restrictive strategy [27] of eliminating COVID-19. A range of controls (policy measures) to stop the transmission of COVID-19 in New Zealand were implemented [28]. In the New Zealand context, elimination (the desired policy outcome) meant that central government was confident chains of transmission in the community had been eliminated for at least 28 days, and that any cases arriving from overseas in the future could be effectively contained [28].

A four-tier alert system was instituted by the central government that mandated policy measures such as: progressively tighter restrictions on people's movement beyond their homes and immediate families, including travelling to work; social distancing and encouraging the wearing of masks outside the home at the higher alert levels; and self-isolating and seeking testing if people felt unwell or experienced symptoms characteristic of COVID-19 infection [26].

On 25 March 2020, New Zealand moved to a Level 4 'lockdown', the highest level of alert, and a National State of Emergency was declared [26]. People were instructed to stay at home except for essential travel for health care or indispensable shopping [8]. Socially distanced recreational activity was allowed in the local area but travel beyond the local area was severely limited [8]. Public gatherings were cancelled, and all public venues closed. Businesses were closed except for essential services such as supermarkets, pharmacies, health clinics, petrol stations and lifeline organisations [8]. All educational facilities were closed [26].

The country progressively moved to lower alert levels: Level 3 towards the end of April and Level 2 in early May 2020 as the spread of the virus slowed and stopped [8]. The lowest level, Alert Level 1, was introduced on 8 June 2020 because community transmission had halted and there were no active cases in the country outside the Managed Isolation and Quarantine facilities (MIQ) [8]. These facilities were established specifically to confine all travelers to New Zealand for 14 days after arrival [8] and, if a traveler tested positive for COVID-19 at any time during the 14 days, they were moved to another quarantine facility for people with COVID-19 [26].

Auckland returned to Alert Level 3, with the rest of the country at Alert Level 2, on 12 August 2020 after four new cases were detected in Auckland [8]. Auckland remained at Alert Level 3 until 30 August, when it moved to Level 2 but with additional restrictions on travel and the size of gatherings [8]. The rest of the country remained at the standard Alert Level 2 until 21 September, when the alert level was downgraded to Level 1 while the

extra restrictions on Auckland residents were relaxed on 21 September, and they returned to Alert Level 1 on 7 October 2020 [28].

The central government commenced a mass vaccination programme for COVID-19 using the Pfizer vaccine, starting with border staff and managed isolation and quarantine workers, in February 2021 [11,29]. The programme was accompanied by an extensive, government-funded publicity campaign using traditional and social media.

## 3. Materials and Methods

### 3.1. Survey Data

Data from two surveys were used in this study. The first survey, the 'Auckland' survey, was of Auckland residents and was conducted over two weeks from 7 September to 22 September 2020 [8]. Auckland was chosen for the survey because it is New Zealand's largest city and is the mostly likely place for community transmission to occur, given the greater number of MIQ facilities and frontline border workers in the city [8]. At the time of the survey, Auckland residents were mostly under Alert Level 2, which meant that they were expected to maintain social distancing when outside their homes and to wear masks in public places. They were also expected to keep track of their movements and to self-isolate and seek testing for COVID-19 if they felt unwell and experienced symptoms associated with COVID-19 [8].

The second survey, the 'regional' survey, was of residents in five regions outside Auckland with, or near, MIQ facilities (Hamilton, Rotorua, Tauranga, Wellington, and Christchurch) and was conducted during the first and second week of March 2021, before the Delta variant was detected in New Zealand and before vaccinations were available to the general public [11]. When the survey commenced, residents in these regions were under Alert Level 2, which meant that they were expected to maintain social distancing when outside their homes and to wear masks in public places [8]. They were also expected to keep track of their movements and to self-isolate and seek testing for COVID-19 if they felt unwell and experienced symptoms associated with COVID-19 [8]. On 8 March 2021 the regions shifted to (the lowest) Alert Level 1, which meant people were expected to wear masks on public transport, were also encouraged to keep track of their movements and to self-isolate and seek testing for COVID-19 if they felt unwell and experienced symptoms associated with COVID-19 [26].

For both surveys, a questionnaire seeking information from the public on their beliefs about, attitudes towards, and willingness to wear face masks, self-isolate and be tested for COVID-19 was designed based on the I$_3$ Compliance Framework [8,20]. The questionnaire is reproduced in Figure S1. Involvement was measured using a condensed version of the Laurent and Kapferer [30] involvement scale (described in Kaine [31]), with respondents rating two statements on each of the five components of involvement (functional, experiential, identity-based, risk-based, and consequence-based). Attitudes were measured using a simple, evaluative scale. The ordering of the statements in the involvement and attitude scales was randomised to avoid bias in responses [8]. Following Kaine et al. [8] respondents indicated their agreement with statements in all the involvement, attitude and belief scales using a five-point rating, ranging from strongly disagree (1) to strongly agree (5).

Respondents' propensity to wear face masks was obtained by asking them if they had worn a face mask when (i) out in public the previous week and (ii) they had gone out to work the previous week [8]. Respondents answered both questions using a five-point scale ranging from 'always' to 'never'. Their propensity to self-isolate was obtained by asking them, 'Thinking about the next few days, would you stay home if you were unwell or had any of the following symptoms: a dry cough, fever, loss of sense of smell, loss of sense of taste, shortness of breath or difficulty breathing?' [8]. We also asked, 'If you were advised to do so by a healthcare professional or public health authority, would you self-isolate for 14 days?' [8]. Both questions were answered using a five-point scale ranging from 'definitely' to 'definitely not'.

Following Kaine et al. [8], information was also sought on the demographic characteristics of respondents, including age, education, and ethnicity, and whether they wore masks, would self-isolate and had been tested for COVID-19. The ethnicity categories were Māori (the Indigenous people of New Zealand), European New Zealander, Pacific Islander, Asian and Other [8].

As with Kaine et al. [8], participation in surveys was voluntary, respondents could leave the survey at any time, and all survey questions were optional and could be skipped. The research approach was reviewed and approved by the Manaaki Whenua–Landcare Research's social ethics process (application no. 2021/10 NK) which is based on the New Zealand Association of Social Science Research code of ethics [8].

The Auckland questionnaire had been piloted with a small random sample of residents (n = 30), and subsequently completed by a larger random sample of residents (n = 1001) who were members of a large-scale, commercial consumer internet panel [8]. The regional questionnaire, which was identical to the Auckland questionnaire, was completed by a large random sample of residents (n = 2000), stratified by regional population, who were also members of a large-scale, commercial consumer internet panel. Panel members receive reward points (which are redeemable for products and services) for completing surveys [8]. For both surveys, an internet link to the questionnaire was distributed to randomly selected members of the panel subject to the constraint that they were resident in the relevant region and were not minors [8]. To reduce respondent fatigue, and as in Kaine et al. [8], a split-sample approach was taken with each respondent answering sets of questions in relation to two of the three measures (mask wearing, self-isolation and getting tested for COVID-19). Measures were randomly allocated among respondents.

It seems reasonable to suppose that virtually all the residents in the regions were aware of COVID-19 at the time we surveyed them as the pandemic had been receiving constant mainstream and social media coverage in New Zealand since February 2020 [8]. It also seems reasonable to suppose that most, if not all, were aware of government claims regarding the social desirability of wearing face masks and social distancing when out in public, and of self-isolating if feeling unwell and getting tested for COVID-19 [8].

Although awareness is a prerequisite for involvement, it does not necessarily equate with involvement. As Kaine et al. [8] observed, widespread awareness of COVID-19 only creates the potential for widespread involvement. The degree to which that potential is realised depends on respondents' beliefs about how COVID-19 could affect the achievement of satisfaction of their needs [8].

For this paper, data were also gathered on the number, dates, and location by District Health Board of COVID-19 cases reported by the New Zealand Ministry of Health [32]. At the time of the second survey there had been less than 2000 cases of COVID-19 in New Zealand, nearly all in Auckland [32]. The data used in the study is available in supplementary Table S1.

As the data on involvement, attitudes, behavioural intentions, and behaviour were collected using the same questionnaire but at different times, a number of confounding factors may give rise to dissimilar results. These are:

- differences in awareness of the prevalence, or nature, of COVID-19;
- differences in the demographic composition of the samples; and
- differences in Alert Level restrictions.

It seems unlikely that differences between the surveys in intentions and behaviour could be the result of differences in awareness of the prevalence, or nature, of COVID-19. Almost 730 cases of COVID-19 were detected in Auckland prior to the Auckland survey [32]. Approximately 570 cases had been detected in the MIQ regions prior to conducting the regional survey [32]. As noted earlier, the pandemic received extensive media coverage, including daily government briefings, in New Zealand during the six months preceding the Auckland survey. The Delta variant of COVID-19, a more highly infectious and severe variant of the virus [33,34], emerged during the five months between the Auckland and regional surveys but was not present in New Zealand at the time the regional survey was

conducted. Given the emergence of the Delta variant, and that the regional survey occurred some five months after the Auckland survey, awareness of COVID-19 among regional respondents could reasonably be expected to be at least as great as it was among Auckland respondents. The potential confounding effects of the demographic composition of the samples and differences in Alert Level restrictions are investigated below.

### 3.2. Methods

The Auckland dataset and the regional (non-Auckland) dataset were combined. Involvement scores were computed for each respondent as the simple arithmetic average of their agreement ratings for the 10 statements in the involvement scales [8]. Attitudes scores were computed as the simple arithmetic average of their agreement ratings for the five statements in the attitude scales [8].

Regional respondents were classified into three categories according to the date when they completed the questionnaire. The first category consisted of respondents who completed the questionnaire prior to 8 March 2021, the date at which Alert Levels changed from 2 to 1. The second category consisted of respondents who completed the questionnaire between 8 March 2021 and 15 March 2021, who could have been reporting on their behaviour under either Alert Level 1 or 2 depending on how they interpreted the phrase 'Did you wear a face mask whenever you went out in public last week?'. The third and final category consisted of respondents who completed the questionnaire after 15 March 2021, and so should have been reporting on their behaviour under Alert Level 1.

The analysis was split into stages as follows. First, statistically significant differences in the demographic composition of the samples were identified using analysis of variance. Second, the effects of differences in the demographic composition of the samples on behavioural intentions and behaviour (wearing of face masks) were identified, again using analysis of variance. Third, statistically significant differences between the two surveys regarding involvement, attitudes, intentions, and behaviour were identified using analysis of variance. Fourth, using the data from both surveys, linear regression analysis was used to predict respondent's intentions and behaviour.

Following Kaine et al. [8,9], we hypothesised that respondents' willingness to take some responsibility for eliminating COVID-19, and their willingness to change normal behaviour, work with others and make sacrifices to eliminate COVID-19 from New Zealand would be a function of their involvement with, and attitude towards, eliminating COVID-19 from New Zealand. We also hypothesised, following Kaine et al. [8,9], that respondents' propensity to self-isolate and wear face masks would be a function of their involvement with, and attitude towards, self-isolating and wearing face masks, respectively.

To account for the possibility that the demographic differences between the surveys might be correlated with relevant omitted variables (e.g., social norms, susceptibility to infection, risk of severe symptoms), respondents' demographic characteristics were also included as predictor variables in the regressions. Dummy variables were created representing respondents' ethnicity with 'Other' treated as the benchmark. Furthermore, two additional dummy variables were included in the regressions, one representing the period when regional respondents were at Alert Level 2 and one representing the first week regional respondents were at Alert Level 1.

We included additional explanatory variables in the regressions for wearing face masks which were intended to account for differences in the frequency with which Auckland and regional respondents wore face masks. We attributed this difference in behaviour to differences in respondents' perceptions of the risk of COVID-19 infection. We hypothesised that respondents would, in some way, use information about the number of cases in their region in forming their perception of the risk of being infected with COVID-19. A positive relationship between reported infection figures and perceived risk of infection seemed plausible; the question was how best the former might be modeled as, probably indirectly, influencing the latter. We explored two alternative specifications in this regard.

To begin with, we assumed that respondents' perception of the risk of infection was proportional to the total number of COVID-19 cases reported in their region prior to the survey, and that their perception of the risk of infection would become increasingly sensitive to the total number of cases, as that total increased. Consequently, two additional variables were included in the regressions predicting the wearing of face masks in public and at work. These were the total number of cases in a respondent's region, and the total number of cases in their region squared (and centred to avoid multi-collinearity). Note that respondents might also use Alert level settings in judging the risk of infection.

Alternatively, respondents' perception of the risk of infection could be proportional to the total number of COVID-19 cases reported in their region prior to the survey, expressed as a fraction of the population of the region, and that their perception of the risk of infection would become increasingly sensitive to the number of cases, expressed as a fraction of the population of the region, as the total number of cases increased. Consequently, an alternative specification for the regressions predicting the wearing of face masks in public and at work was to include the total number of cases in a respondent's region expressed as a proportion of the regional population, and the total number of cases expressed as a proportion of the regional population squared (and centred to avoid multi-collinearity) as predictor variables. Note again that respondents might also use Alert level settings in judging the risk of infection.

Statistical analyses were conducted using IBM SPSS Statistics (v28, Windows) [35]. Given the size of the sample we set the level of statistical significance at $p < 0.01$ to ensure we only interpreted associations that were both statistically significant and meaningful [36].

## 4. Results

To begin with, although the samples were broadly similar with regard to their age, education, and income composition, they differed substantially with respect to gender and ethnicity (see Table 1). Approximately 53% of respondents to the Auckland survey were women whereas approximately 65% of respondents to the regional survey were women.

There were statistically significant but weak associations [36] between the demographic characteristics and willingness to take responsibility for eliminating COVID-19 and willingness to change normal behaviour, make sacrifices and work with others to eliminate COVID-19 (see Table 2). There were also some statistically significant but weak associations [34] between demographic characteristics and the wearing of face masks and willingness to self-isolate (see Table 3). These associations may simply be attributable to the size of the samples. Nevertheless, differences in demographic characteristics such as age, ethnicity, gender, and education may influence perceptions of the danger to health posed by COVID-19. This suggests that differences in the demographic composition of the two surveys could partly explain differences in intentions and behaviour in the two surveys.

We found a statistically significant, but extremely small, difference between respondents in the Alert level categories in the mean scores for willingness to change normal behaviour ($\eta^2 = 0.007$), make sacrifices ($\eta^2 = 0.006$) and work with others to eliminate COVID-19 ($\eta^2 = 0.007$). We also found a statistically significant, but extremely small, difference between respondents in the different Alert level categories with respect to mean scores for wearing face masks in public ($\eta^2 = 0.012$) and wearing face masks at work ($\eta^2 = 0.011$). This suggests that the difference in Alert Levels between the two surveys may also partly explain differences in intentions and behaviour in the two surveys.

**Table 1.** Age, education, ethnicity and income distribution of respondents.

| Age Category (Years) | Percentage of Auckland Respondents | Percentage of Regional Respondents |
|---|---|---|
| 18–29 | 22.8 | 13.2 |
| 30–39 | 21.8 | 22.6 |
| 40–49 | 18.4 | 21.5 |
| 50–59 | 13.1 | 12.5 |
| 60–69 | 12.5 | 13.8 |
| 70 and over | 11.4 | 16.4 |
| **Education Category** | | |
| Some or all of secondary school | 14.2 | 19.5 |
| Certificate (1–6) | 12.4 | 19.1 |
| Diploma (5–7) | 14.3 | 17.5 |
| Bachelor | 33.6 | 23.4 |
| Post-graduate diploma/certificate | 10.2 | 11.1 |
| Post-graduate degree | 15.3 | 9.4 |
| **Ethnic Category** | | |
| European | 53.3 | 72.1 |
| Māori | 4.4 | 13.5 |
| Pacific Islander | 4.7 | 1.8 |
| Other | 37.6 | 12.7 |
| **Income Category** | | |
| Less than $20,000 | 4.3 | 8.5 |
| $20,000 to $50,000 | 21.2 | 26.0 |
| $50,000 to $70,000 | 18.6 | 21.7 |
| $70,000 to $100,000 | 22.0 | 22.1 |
| More than $100,000 | 33.8 | 21.6 |

**Table 2.** Demographics and behavioural intentions.

| Characteristic | Feel Some Responsibility | Change Normal Behaviour | Work with Others | Make Sacrifices |
|---|---|---|---|---|
| Age | 0.006 * | 0.007 * | 0.019 * | 0.014 * |
| Gender | 0.003 * | 0.005 * | 0.003 * | 0.005 |
| Education | 0.009 * | 0.006 * | 0.008 * | 0.006 * |
| Ethnicity | 0.006 * | 0.003 | 0.003 | 0.001 |
| Income | 0.016 * | 0.014 * | 0.009 * | 0.011 * |

Values are $\eta^2$, the proportion of the variance in intention explained by the variance in the socio-demographic variable [36]. For example, differences in income explain 1.4% of the variation in intention to change normal behaviour ($\eta^2 = 0.014$). * Anova F-test indicates statistically significant effect ($p < 0.01$).

**Table 3.** Demographics, wearing face masks and intention to self-isolate.

| Characteristic | Wore a Face Mask in Public | Wore a Face Mask at Work | Stay Home If Unwell | Stay Home If Instructed by Health Authority |
|---|---|---|---|---|
| Age | 0.011 * | 0.016 * | 0.039 * | 0.029 * |
| Gender | 0.009 * | 0.016 * | 0.007 * | 0.004 |
| Education | 0.021 * | 0.029 * | 0.004 | 0.003 |
| Ethnicity | 0.052 * | 0.066 * | 0.009 * | 0.015 * |
| Income | 0.002 | 0.005 | 0.010 * | 0.014 * |

Values are $\eta^2$ [36]. * Anova F-test indicates statistically significant effect ($p < 0.01$).

*4.1. Intentions and Behaviour*

The purpose of this analysis was to explain differences in the propensity of Auckland and regional respondents to comply with wearing face masks in public and at work,

given that their behavioural intentions were similar. Note that satisfactory reliabilities [37] were obtained in both surveys for the involvement and attitudinal scales with respect to eliminating COVID-19, wearing face masks, self-isolating when unwell and getting tested for COVID-19 (see Table 4).

**Table 4.** Reliability of involvement and attitude scales.

| | Auckland | Regional |
|---|---|---|
| Involvement with eliminating COVID-19 | 0.847 | 0.853 |
| Involvement with wearing face masks | 0.852 | 0.863 |
| Attitude towards wearing face masks | 0.854 | 0.814 |
| Involvement with self-isolating when unwell | 0.707 | 0.758 |
| Attitude towards self-isolating when unwell | 0.795 | 0.800 |
| Involvement with getting tested for COVID-19 | 0.821 | 0.807 |
| Attitude towards getting tested for COVID-19 | 0.849 | 0.813 |

Values are Cronbach's alpha [37].

The means for all the involvement, attitude, intention, and behaviour variables for both surveys are reported in Table 5. Where the means for the two surveys were statistically significantly different, the magnitude of the differences, as measured by effect size [34], is also reported in the table. An inspection of the table reveals that wearing face masks in public and at work were the only variables for which the means were statistically significantly different, and the magnitude of the difference was large ($\eta^2 > 0.05$), for the two surveys.

These results suggest that, on average, respondents in both surveys were similar in their motivations (as measured by involvement) and attitudes towards eliminating COVID-19, wearing face masks, self-isolating and getting tested for COVID-19. They were similar, on average, regarding their intentions to take some responsibility for eliminating COVID-19, and their intentions to change their normal behaviour, work with others and make sacrifices to eliminate COVID-19 from New Zealand. They were also similar, on average, regarding their intentions to self-isolate and get tested for COVID-19. Relatedly, Auckland and regional respondents were similar in their patterns of beliefs about COVID-19, eliminating COVID-19 from New Zealand, and the advantages and disadvantages of wearing face masks, self-isolating and getting tested for COVID-19 [9].

The only substantive difference between the two samples relates to the wearing of face masks, with the means for the regional sample being substantially lower than the means for the Auckland sample.

### 4.2. Predicting Intentions and Behaviour

The explanatory power of the regressions predicting respondents' behavioural intentions, and the resulting parameter estimates, are reported in Table 6. The regressions were statistically significant and, for cross-sectional data, a substantial proportion of the variance in respondents' intentions was explained by their involvement with, and attitude towards, eliminating COVID-19. The results show that involvement and attitude account for the bulk of the explained variation in the dependent variables. The variation in respondents' intentions were only weakly related, if at all, to their demographic characteristics or, for regional respondents, changes in Alert Levels.

**Table 5.** Involvement, attitude, intentions, and behaviour.

| | Auckland [1] | Regional [1] | Effect Size [2] |
|---|---|---|---|
| Involvement with eliminating COVID-19 * | 3.90 | 3.83 | 0.003 |
| Attitude towards eliminating COVID-19 | 4.19 | 4.19 | - |
| Feel some responsibility * | 4.04 | 3.94 | 0.003 |
| Change normal behaviour * | 4.22 | 4.11 | 0.003 |
| Work with others * | 4.34 | 4.25 | 0.003 |
| Make sacrifices | 4.13 | 4.10 | - |
| Involvement with wearing face masks * | 3.51 | 3.40 | 0.005 |
| Attitude towards wearing face masks * | 4.08 | 3.85 | 0.016 |
| Wore a face mask in public * | 3.93 | 2.54 | 0.195 |
| Wore a face mask at work * | 3.62 | 2.64 | 0.082 |
| Involvement with self-isolating when unwell | 3.73 | 3.68 | - |
| Attitude towards self-isolating when unwell * | 4.39 | 4.24 | 0.010 |
| Stay home if unwell | 1.48 | 1.50 | - |
| Stay home if instructed by health authority | 1.32 | 1.27 | - |
| Involvement with getting tested for COVID-19 | 3.41 | 3.43 | - |
| Attitude towards getting tested for COVID-19 | 4.15 | 4.08 | - |

[1] Values for Auckland and regional are mean scores. [2] Values are $\eta 2$ [36]. * Anova F-test indicates significant differences in means ($p < 0.01$).

The number of observations was 2586 for each regression. Willingness to self-isolate, when unwell or instructed to do so by a health authority, was strongly and positively influenced by involvement with, and attitudes towards, self-isolating (see Table 7). As with the other behavioural intention variables, the variation in respondents' intentions to self-isolate were only weakly related, if at all, to their demographic characteristics or, for regional respondents, changes in Alert Levels.

The results of the regressions predicting the frequency of wearing face masks in public and at work, with and without the additional risk perception variables, are reported in Tables 8 and 9.

**Table 6.** Standardised parameter estimates for behavioural intentions.

| | Feel Responsible for Eliminating COVID-19 | Prepared to Change Normal Behaviour | Willing to Make Sacrifices | Willing to Work Together |
|---|---|---|---|---|
| Involvement with eliminating COVID-19 | 0.466 ($p < 0.001$) | 0.479 ($p < 0.001$) | 0.399 ($p < 0.001$) | 0.469 ($p < 0.001$) |
| Attitude towards eliminating COVID-19 | 0.232 ($p < 0.001$) | 0.272 ($p < 0.001$) | 0.353 ($p < 0.001$) | 0.295 ($p < 0.001$) |
| Age | 0.044 (($p = 0.007$) | 0.044 (($p = 0.005$) | 0.062 ($p < 0.001$) | 0.058 ($p < 0.001$) |
| Gender | 0.037 ($p = 0.017$) | 0.047 ($p = 0.001$) | 0.023 (($p = 0.118$) | 0.052 ($p < 0.001$) |
| Education | 0.040 ($p = 0.011$) | −0.008 ($p = 0.593$) | −0.015 ($p = 0.333$) | 0.002 ($p = 0.916$) |
| Income | 0.081 ($p < 0.001$) | 0.034 ($p = 0.027$) | 0.004 ($p = 0.817$) | 0.045 ($p = 0.003$) |
| European NZ | −0.049 ($p = 0.009$) | −0.020 ($p = 0.272$) | −0.018 ($p = 0.315$) | 0.000 ($p = 0.984$) |
| Māori | −0.051 ($p = 0.004$) | −0.042 ($p = 0.014$) | −0.053 ($p = 0.002$) | −0.023 ($p = 0.166$) |
| Pacific Islander | −0.004 ($p = 0.801$) | −0.009 ($p = 0.566$) | −0.016 ($p = 0.281$) | 0.019 ($p = 0.187$) |
| Regions at Alert Level 2 | −0.013 ($p = 0.386$) | −0.040 ($p = 0.007$) | −0.002 ($p = 0.875$) | 0.001 ($p = 0.938$) |
| First week regions at Alert Level 1 | −0.012 ($p = 0.453$) | 0.000 ($p = 0.996$) | −0.003 ($p = 0.842$) | 0.015 ($p = 0.320$) |
| Adjusted R$^2$ | 0.43 | 0.48 | 0.48 | 0.50 |
| F-Test significance | <0.001 | <0.001 | <0.001 | <0.01 |

**Table 7.** Standardised parameter estimates for intention to self-isolate.

| | Intending to Stay Home if Feeling Unwell (n = 1697) | Stay Home if Instructed to Do So (n = 1697) |
|---|---|---|
| Involvement with self-isolating | 0.131 ($p < 0.001$) | 0.100 ($p < 0.001$) |
| Attitude towards self-isolating | 0.330 ($p < 0.001$) | 0.399 ($p < 0.001$) |
| Age | 0.174 ($p < 0.001$) | 0.090 ($p < 0.001$) |
| Gender | 0.086 ($p < 0.001$) | 0.041 ($p = 0.072$) |
| Education | 0.038 ($p = 0.103$) | 0.001 ($p = 0.958$) |
| Income | 0.043 ($p = 0.058$) | 0.036 ($p = 0.115$) |
| European NZ | 0.038 ($p = 0.159$) | 0.016 ($p = 0.556$) |
| Māori | 0.009 ($p = 0.718$) | −0.032 ($p = 0.214$) |
| Pacific Islander | 0.032 ($p = 0.158$) | −0.016 ($p = 0.486$) |
| Regions at Alert Level 2 | −0.036 ($p = 0.112$) | −0.055 ($p = 0.014$) |
| First week regions at Alert Level 1 | −0.029 ($p = 0.209$) | 0.017 ($p = 0.441$) |
| Adjusted $R^2$ | 0.21 | 0.23 |
| F-Test significance | <0.001 | <0.001 |

Differences in sample size between Tables 7–9 are the result of split sampling.

The regressions where the perception of the risk of infection was assumed to be proportional to the total number of COVID-19 cases were superior to the regressions where the perception of the risk of infection was assumed to be proportional to the total number of COVID-19 cases expressed as a fraction of the population of the region.

Willingness to wear face masks in public was strongly and positively influenced by involvement with, and attitudes towards, wearing face masks. As was the case with behavioural intentions, the variation in respondents' wearing of face masks in public was only weakly related to their demographic characteristics. Changes in Alert Levels did appear to have some influence on the wearing of face masks in public by regional respondents, though this influence was weakened when the variables intended to account for respondents' perception of the risk of COVID-19 infection were included in the regression. As hypothesised, the cumulative number of COVID-19 cases in a region had a positive influence on the wearing of face masks in public, and this influence strengthened as case numbers increased.

Willingness to wear face masks at work was positively influenced by involvement with, and attitudes towards, wearing face masks although the effects of attitude are weaker than was the case for wearing face masks in public. The variation in respondents' wearing of face masks at work was related to gender, education and, slightly, ethnicity. Changes in Alert Levels did appear to influence the wearing of face masks at work by regional respondents. The variables intended to account for respondents' perception of the risk of COVID-19 infection had a much smaller influence on mask wearing at work compared to their influence on wearing face masks when out in public, perhaps reflecting the lower degree of autonomy in decision-making that individuals have in the workplace.

**Table 8.** Standardised parameter estimates for wearing face masks in public.

| | Wore a Face Mask in Public (n = 1650) | Wore a Face Mask in Public (n = 1650) | Wore a Face Mask in Public (n = 1650) |
|---|---|---|---|
| Involvement with wearing face masks | 0.228 ($p < 0.001$) | 0.246 ($p < 0.001$) | 0.230 ($p < 0.001$) |
| Attitude towards face masks | 0.235 ($p < 0.001$) | 0.198 ($p < 0.001$) | 0.227 ($p < 0.001$) |
| Total number of cases | | 0.185 ($p < 0.001$) | |
| Total cases squared | | 0.140 ($p < 0.001$) | |
| Total number of cases as a proportion of the regional population | | | 0.107 ($p < 0.001$) |
| Total cases as a proportion of the regional population squared | | | 0.047 ($p = 0.031$) |
| Age | −0.067 ($p = 0.003$) | −0.064 ($p = 0.003$) | −0.063 ($p = 0.005$) |
| Gender | −0.103 ($p < 0.001$) | −0.082 ($p < 0.001$) | −0.098 ($p < 0.001$) |
| Education | 0.065 ($p = 0.003$) | 0.060 ($p = 0.004$) | 0.061 ($p = 0.005$) |
| Income | −0.030 ($p = 0.176$) | −0.035 ($p = 0.094$) | −0.032 ($p = 0.139$) |
| European NZ | −0.074 ($p < 0.004$) | −0.011 ($p = 0.660$) | −0.063 ($p = 0.016$) |
| Māori | 0.063 ($p = 0.010$) | 0.120 ($p < 0.001$) | 0.072 ($p = 0.003$) |
| Pacific Islander | 0.010 ($p = 0.636$) | 0.011 ($p = 0.600$) | 0.010 ($p = 0.637$) |
| Regions at Alert Level 2 | −0.224 ($p < 0.001$) | −0.105 ($p < 0.001$) | −0.199 ($p < 0.001$) |
| First week regions at Alert Level 1 | −0.258 ($p < 0.001$) | −0.097 ($p < 0.001$) | −0.224 ($p < 0.001$) |
| Adjusted $R^2$ | 0.33 | 0.38 | 0.33 |
| F-Test significance | <0.001 | <0.001 | <0.001 |

Differences in sample size between Tables 7–9 are the result of split sampling.

These results support our hypothesis that the difference between Auckland and regional respondents in the wearing of face masks can be attributed to differences in perceptions of the risk of infection. If regional respondents did perceive the risk of infection from COVID-19 to be lower than Auckland respondents, then they should be less likely than Auckland respondents to seek testing for COVID-19 unless they felt unwell. That is to say, the proportion of respondents who felt unwell when they were tested for COVID-19 should be significantly higher among regional respondents than Auckland respondents. We found this to be the case with approximately 69.7% of regional respondents feeling unwell when they were tested for COVID-19 compared to 50.4% of Auckland respondents ($p < 0.001$).

**Table 9.** Standardised parameter estimates for wearing face masks at work.

| | Wore a Face Mask at Work (n = 1225) | Wore a Face Mask at Work (n = 1225) | Wore a Face Mask at Work (n = 1225) |
|---|---|---|---|
| Involvement with wearing face masks | 0.288 ($p < 0.001$) | 0.298 ($p < 0.001$) | 0.288 ($p < 0.001$) |
| Attitude towards face masks | 0.089 ($p = 0.003$) | 0.073 ($p = 0.017$) | 0.089 ($p = 0.004$) |
| Total number of cases | | 0.075 ($p = 0.077$) | |
| Total cases squared | | 0.102 ($p = 0.008$) | |
| Total number of cases as a proportion of the regional population | | | 0.068 ($p = 0.023$) |
| Total cases as a proportion of the regional population squared | | | 0.059 ($p = 0.030$) |
| Age | −0.080 ($p = 0.002$) | −0.079 ($p = 0.002$) | −0.078 ($p = 0.003$) |
| Gender | −0.135 ($p < 0.001$) | −0.126 ($p < 0.001$) | −0.135 ($p < 0.001$) |
| Education | 0.108 ($p < 0.001$) | 0.107 ($p < 0.001$) | 0.108 ($p < 0.001$) |
| Income | −0.066 ($p = 0.013$) | −0.069 ($p = 0.009$) | −0.068 ($p = 0.010$) |
| European NZ | −0.059 ($p = 0.061$) | −0.018 ($p = 0.585$) | −0.050 ($p = 0.112$) |
| Māori | 0.127 ($p < 0.001$) | 0.158 ($p < 0.001$) | 0.128 ($p < 0.001$) |
| Pacific Islander | 0.043 ($p = 0.096$) | 0.043 ($p = 0.094$) | 0.044 ($p = 0.092$) |
| Regions at Alert Level 2 | −0.135 ($p < 0.001$) | −0.076 ($p = 0.007$) | −0.121 ($p < 0.001$) |
| First week regions at Alert Level 1 | −0.168 ($p < 0.001$) | −0.088 ($p = 0.004$) | −0.149 ($p < 0.001$) |
| Adjusted $R^2$ | 0.26 | 0.27 | 0.26 |
| F-Test significance | <0.001 | <0.001 | <0.001 |

Differences in sample size between Tables 7–9 are the result of split sampling. Differences in sample size between Tables 8 and 9 result from some respondents did not leave home to work.

## 5. Discussion

These results suggest that, on average, respondents in both surveys were similar in their beliefs about, attitudes towards, and motivations regarding eliminating COVID-19, wearing face masks, self-isolating and getting tested for COVID-19. They were also similar, on average, regarding their behavioural intentions. The only substantive difference between the two samples relates to the wearing of face masks, with the means for the regional sample being substantially lower than the means for the Auckland sample. The difference, on average, roughly corresponds to regional respondents reporting that they only wore face masks sometimes (at best) when they were out in public whereas Auckland respondents reported they often wore face masks when they were out in public.

### 5.1. Implications

The intention of people to adopt behaviours to prevent the spread of COVID-19 such as wearing face masks, self-isolating and getting tested for COVID-19 has been the subject of numerous studies [8,9,38–45] which have found that people's intentions to adopt these behaviours depends on their beliefs about, and attitudes towards, them. Accordingly, many of these studies recommend investing in promotion to change beliefs about, and so

attitudes towards, preventative behaviours to increase adoption. Our results have three important implications for such recommendations.

The first concerns the fact that these findings are a reminder that intentions [46–49] do not always immediately, or inevitably, translate into actions. While it is undoubtedly true that changing attitudes can change behaviour, promotional efforts intended to change beliefs and attitudes about preventative behaviours are unlikely to meet with complete success unless health authorities also seek to identify the factors that:

- trigger the translation of intentions into actions, and
- prevent those who are intending to act from acting.

For example, with respect to wearing face masks our results suggest that respondents to our surveys may have relied on the number of COVID-19 infections in their region, together with changes in Alert levels, as signals to trigger the translation of intention into action with respect to self-protection [48,50]. This may help to explain location-related differences in the wearing of face masks [51] and means that providing timely and easily accessible information on the number of infections resulting from community transmission in each region is important. Relatedly, it also means that providing timely and easily understandable information on the danger to health posed by different variants of COVID-19 is essential if the public is to set a reasonable criterion of the number of infections from community transmission that they should observe to trigger action. This is supposing that there is a relationship, most likely an inverse relationship, between the seriousness of the health risk posed by a variant and the threshold for infections by community transmission below which intentions remain just that, intentions.

Second, bearing in mind the difficulties of engaging with those who are not involved and getting them to observe measures to prevent or slow the spread of COVID-19 [8], our results reinforce the importance of ensuring that measures to prevent the spread of COVID-19 are as simple and convenient to adopt as possible [52]. In other words, improving the ease and convenience of adopting preventative behaviours may, in fact, be more effective in changing behaviour than promotional efforts aimed at changing people's beliefs and attitudes [8]. In the context of measures to prevent the spread of COVID-19, this means ensuring that compliance requires little effort and thought, and is as stress-free as possible [25,38,53].

People's willingness to wear face masks depends on their beliefs about the effectiveness of face masks in protecting them from infection [8,54]. Consequently, developing and promoting to the public clear guidelines on wearing face masks and increasing promotional efforts dispelling myths about the efficacy of masks are important strategies for encouraging the wearing of face masks [47]. While such strategies may meet with success in encouraging intentions, they are only part of the story. Understanding public perception of the personal risk of COVID−19 infection is also fundamental for establishing effective prevention measures [55,56]. Consequently, if information on case numbers influences public perceptions of risk, then providing timely, trustworthy, and easily accessible regional information on the number and incidence of infections resulting from community transmission is critical. While there are numerous studies of risk perception with respect to COVID-19 [56–60], we were unable to identify any studies that had directly investigated the association, if any, between the incidence of COVID-19 infections and public perceptions of the risk of exposure to COVID-19.

Third, it is important to bear in mind that, for New Zealanders at least, wearing a face mask when out in public means constantly disrupting routine behaviours. Consequently, wearing a face mask requires much more time and effort than getting vaccinated for COVID-19, a non-routine action that, currently, only needs to be performed up to four times. This suggests that the perceived risk of infection that triggers the translation of the intention to wear a face mask into action is likely to be higher (ceteris paribus) than that required to trigger activation of the intention to get vaccinated for COVID-19.

Relatedly, this suggests that, once the number of infections by community transmission begins to decline, the public will desire an end to measures to prevent the spread of COVID-19

that constantly disrupt routine behaviours far earlier than for other, less unsettling measures. This means, for instance, that the observance of the measures mask wearing, social distancing, using tracer apps and showing vaccine passes is likely to deteriorate even though measures such as being vaccinated and tested are still strongly supported. Ironically, success in vaccinating the public may well encourage the faster abandonment of mask wearing and social distancing if vaccinations are perceived to reduce the risk of transmission and the severity of symptoms. How acceptable this outcome is will depend on the protection vaccination actually provides, and its persistence.

This raises the possibility that, because misinformation can undermine compliance with COVID-19 measures [61,62], the distribution of misinformation about the health risks posed by COVID-19 through social media could lead to measures that disrupt daily routines being adopted more slowly, and abandoned more rapidly, than is desirable. Misinformation may provide a self-serving rationale for not complying with measures that require persistent effort. This reinforces the importance of providing timely, accurate and trustworthy regional information on the spread of COVID-19 variants by community transmission and the severity of the health dangers posed by each variant. When it comes to investing resources to combat misinformation about COVID-19, these considerations suggest that government authorities must be mindful of the relative importance of combating misinformation that targets people's beliefs and attitudes and that which targets triggers to action.

### 5.2. Limitations and Areas for Future Research

Our findings are subject to a number of qualifications. First, as the survey samples were drawn from internet-based consumer panels, there may be selection bias. While the extent of this bias is unknown, it does seem reasonable to suppose that people with low-to-mild involvement may be under-represented in the sample.

Second, as we depended on self-reporting to measure the wearing of face masks, willingness to self-isolate and being tested for COVID-19, social desirability bias [63,64] may have affected measurements of these behaviours. However, while there may be a correlation between intensity of involvement and social desirability bias, the dramatic difference between Auckland and regional respondents in their self-reported frequency of wearing face masks suggests that the degree of social desirability bias in our study is small. See Kaine et al. [6] for a more detailed discussion of this matter.

Third, the adoption of behaviours such as the wearing of face masks has been associated with a range of variables including feelings of stress in relation to COVID-19 [23]. We did not include such variables in our analysis and, while the correlation between these variables and involvement is unknown, it is likely to be positive.

Fourth, the adoption of preventative behaviours such as the wearing of face masks and social distancing has been associated with a range of psychological traits such as pro-sociability and empathy [65–67]. The correlation between these psychological traits and involvement deserves further study, as does the direction of causation between them.

Finally, the extent to which our results and findings generalise to other countries and epidemics is unknown. The potential for differences in involvement to explain differences in compliance with public health measures in other settings is worth investigating. As are differences in perceptions of the risk of infection, and the cues used by the public to infer such risks.

### 6. Conclusions

Governments are seeking to slow the spread of COVID-19 by implementing measures that encourage, or mandate, changes in people's behaviour. The success of these measures depends on (1) the willingness of individuals to change their behaviour and (2) their commitment and capacity to translate that intention into actions. Consequently, understanding and predicting the willingness of individuals to change their behaviour, and their enthusiasm to act on that willingness, is critical in assessing the likely effectiveness of these measures in slowing the spread of the virus.

Our key result was that, while respondents in both surveys had similar beliefs, attitudes, motivations, and behavioural intentions regarding eliminating COVID-19, wearing face masks, self-isolating and getting tested for COVID-19, they differed radically in the frequency with which they wore face masks when they were out in public. This difference was attributable to differences in perceptions of the risk of COVID infection (as measured by the number of COVID cases). This result clearly shows that intentions do not necessarily translate into actions and that efforts to change behaviour, by seeking to change beliefs and attitudes, can be misplaced if the factors that influence the translation of intentions into action are ignored.

**Supplementary Materials:** The following supporting information can be downloaded at: https://www.mdpi.com/article/10.3390/covid2110109/s1, Figure S1: Questionnaire; Table S1: Data.

**Author Contributions:** Conceptualization, G.K., V.W. and S.G.; methodology, G.K. and V.W.; formal analysis, G.K..; investigation, G.K. and V.W.; resources, S.G.; data curation, G.K.; writing—original draft preparation, G.K., V.W. and S.G.; writing—review and editing, G.K., V.W. and S.G.; project administration, G.K. and S.G.; funding acquisition, S.G. All authors have read and agreed to the published version of the manuscript.

**Funding:** This research was funded by the New Zealand Ministry for Business, Innovation and Employment (https://www.mbie.govt.nz/, accessed on 23 October 2022) through the Te Pūnaha Matatini—NZ COVID Modelling Programme (https://www.tepunahamatatini.ac.nz/, accessed on 23 October 2022). MWLR Client project number: UOAX1941.

**Institutional Review Board Statement:** The study was conducted in accordance with the Declaration of Helsinki and approved by the Ethics Committee of Manaaki Whenua–Landcare Research (protocol code 2021/10 NK, 27 January 2021).

**Informed Consent Statement:** Informed consent was obtained from all subjects involved in the study.

**Data Availability Statement:** The data presented in this study are available in Supplementary Materiala, Table S1.

**Acknowledgments:** We would sincerely like to thank those panelists throughout New Zealand who completed our questionnaires. Thanks also to our referees for their time, patience, constructive advice.

**Conflicts of Interest:** The authors declare no conflict of interest. The funders had no role in the design of the study; in the collection, analyses, or interpretation of data; in the writing of the manuscript; or in the decision to publish the results.

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
