# Peer review of "Motivation, Intention and Action: Wearing Masks to Prevent the Spread of COVID-19"

_covid, doi:10.3390/covid2110109_

Round 1
Reviewer 1 Report
Lines 52-56
2. Materials and Methods
2.1 Theory
Research and analysis focused on the process underlying decisions that direct the actions of individuals has long included recognition that there are two phases to the process
A) Theory section. While the authors distinguish behavioural intention and action intention, it is not clear how this explains behaviors during the pandemic. For example, many intend to lose weight by going to a fitness center, but how does this explain specific behaviors like wearing a mask during a pandemic? COVID-19 was a traumatic and unique time in human history that resulted in many changes. For example, people moved after a loss of a job or loved one, which is much different from a want. The authors describe the work from Bagozzi (2006) on their model of “implementation of behavioural intentions”, but the logic is tenuous. The explanation on “barriers” to wearing a mask explain why people might not wear a mask. However, it does not explicitly explain why people intended to wear a mask.
B) Furthermore, it is hard to follow the interrelationship between the results and how it builds to the theory. The theory is too broad to explain the phenomenon of mask wearing intentions in a pandemic in the way it is currently written in the manuscript.
C) Moreover, a key finding is the relationship between number of cases in the region and intentions to wear a mask. There is not a clear part in the theory or explanation for why this may have occurred. Lines 479-486 draws some connections, but it is based on assumptions. It assumed people are closely watching COVID-19 Alerts of cases and responding accordingly to it. The authors need to better delineate the difference people cared about COVID-19 news when their first (September 2020) and second (March 2021) survey to now. Many people developed COVID-19 news fatigue. In 2020, ask the common person the number of COVID-19 cases and it was likely they knew because of salience. Ask the common person today and they likely do not know the infection case numbers in the country. Please make this distinction or explain how New Zealanders were attentive to the four-tier alerts. This relates to the logic of lines 310-321 because the authors connect potential results (in the method section) to later variants of COVID-19. How closely are New Zealanders still following the COVID-19 Alert system?
Lines 310-321
Furthermore, the cases of COVID-19 in New Zealand were relatively small compared to other industrialized nations. New Zealanders cooperated with mandates and recommended public health policies. They may have a greater sense of civic duty or conscientiousness to follow health guidelines that could explain why alert level associated with intent to wearing a mask in public. This connection is not clearly explained. How do we know participants were aware of the current alert level? Perhaps mentioning the cooperation/communal response of New Zealanders will help delineate why they were successful in reducing spread somewhere in the discussion.
Method
This manuscript references the previous work done by the researchers as reference [8].
Kaine G, Greenhalgh S, Wright V, Compliance with Covid-19 measures: Evidence from New Zealand. PLoS ONE 2022; 17(2): e0263376. https://doi.org/10.1371/journal.pone.0263376
Lines 99-101
In two studies [8, 9] in different regions, behavioural intentions with respect to wearing face masks, self-isolating and getting tested for COVID-19 were investigated together with self-reports of actual behaviour with respect to wearing face masks and getting tested.
Lines 208-218
Data from two surveys were used in this study. The first survey, the ‘Auckland’ survey, was of Auckland residents and was conducted over two weeks from 7 September to 22 September 2020 [8]. Auckland was chosen for the survey because it is New Zealand’s largest city and is the mostly likely place for community transmission to occur, given the greater number of MIQ facilities and frontline border workers in the city [8]. At the time of the survey, Auckland residents were mostly under Alert Level 2, which meant that they were expected to maintain social distancing when outside their homes and to wear masks in public places. They were also expected to keep track of their movements and to self-isolate and seek testing for COVID-19 if they felt unwell and experienced symptoms associated with COVID-19 [8].
Lines 220-294 reference the article [8] as well.
D) Please explain, is this the same data analyzed from the previously published work? Please specifically address how this data different or the same?
Lines 433-438
Also, following Kaine et al. [8, 9], we hypothesised that respondents’ propensity to self-isolate and wear face masks would be a function of their involvement with, and attitude towards, self-isolating and wearing face masks, respectively. Consequently, we estimated regressions with respondents’ self-reported willingness to self-isolate and frequency of wearing of face masks as dependent variables.
E) This hypothesis is redundant and wordy in its logic. Please revise. What is the purpose of this hypothesis? To replicate previous findings as a baseline?
Lines 555-558
F) Tables 1 and 2 show “data” “title” and “entry 1” etc. in the tables. This appear to be a mistake. There is a Table 1 and 2 earlier in the manuscript.
G) Please state which version of SPSS was used. Which regression procedure was used?
Discussion:
H) The discussion needs to delineate sections (e.g., limitations, implications, future research). The authors start paragraphs with second and third and a few paragraphs later second and third. Create more sections to help the readers quickly follow along.
Lines 610-612
I) Covid-19 is written with lower case words. Other parts of the paper state COVID-19 with all capitalizations. Please change for consistency
J) New Zealand is a unique country that did a phenomenal job reducing early infections during the start of the COVID-19 pandemic. The authors need to create a limitations paragraph that better delineate the generalizability of results to other nations. For example, New Zealand is a democratic nation that had mandates, but not complete city lockdowns like in China. It was not primarily recommending policies either like many cities in the United States. How did mandates versus recommendations factor into preventative behaviours?
Reviewer 2 Report
In this paper, the authors analyzed two regional surveys to study the motivation, intention and action related to the wearing masks as preventive device to limit the spread of COVID-19. The first survey was conducted between September 07 and 22, 2020. The second one was conducted during the first and second week of March 2021.
In my opinion, a summary of the questions reported within each survey is useful to better understand their informative content. Furthermore, in “Materials and Methods” the methods are missing… some information may be derived from section 2.3, but the methodology applied for the analysis has to be comprehensively reported. The lines 334-344 refer to methodology.
Furthermore, the methodology description will have to describe the statistical method applied to evaluate the p-value and the threshold used for this one (I hypothesized 0.05). In addition, the line 325 should be moved into methods “Statistical analyses were conducted using SPSS”, by indicating version and operating system. To give an example, “IBM SPSS Statistics (v.25, Windows)”
I suggest moving sections 2.1 and 2.2 into a separate section; these do not refer to the methodology or materials, but these report useful information to understand the topic. To give an example, a section, entitled “Background”, could be reported after the introduction. Furthermore, I suggest reporting some line of introduction related to the COVID-19 trend, for instance authors could cite the study (PMID:35885152) related to epidemiology of Covid-19 in several countries (USA, Italy, France, Sweden, UK), the reference is reported as follows: https://pubmed.ncbi.nlm.nih.gov/35885152
The lines 364-382 are a discussion of results. I suggest moving into Section 4.
At line 562 authors report a lot of references related to the topic. I suggest comment some of these, as well as moving this line into “Background”. In this context, I suggest also similar other articles on Pubmed: PMID:33809403, PMID:33797377.
Otherwise, Section 5 (“Conclusions”) is poor, authors should extend it. In my opinion, a section for conclusions should summarize the key points of the manuscript, by reporting also the essential information related to the results obtained during the test and discussion. The authors could explain very briefly what has been done and what has been demonstrated, confirming the importance of the paper.
I think that this manuscript is very interesting; however, the mentioned majors should be addressed.
Minors:
- The manuscript contains typos and grammar mistakes.
- Authors have to remove the example table proposed by the template… for instance, Table1 at line 555, Table 2 at line 557. Similarly, authors have to remove unused information of the template.
Round 2
Reviewer 1 Report
The authors did not address reviewer comments regarding specifying a theory and how results build on this theory.
The COVID-19 pandemic altered society is many ways, such as creating an Alert system which can cause fear to promote a certain behavior. How does behavioral intention address how people behave in crisis or major shifts?
Please review previous comments and address.
Author Response
The authors did not address reviewer comments regarding specifying a theory and how results build on this theory. The COVID-19 pandemic altered society is many ways, such as creating an Alert system which can cause fear to promote a certain behavior. How does behavioral intention address how people behave in crisis or major shifts? Please review previous comments and address.
We have revised the theory section again to make more explicit:
- That the theory underpinning models of behavioural intentions applies to novel situations requiring non-routine decision-making (lines 62-72 and 101-123)
- That behavioural intentions interact with implementation factors such as perception of infection risk to influence wearing a mask, as well as not wearing a mask (lines 201-204).
We already acknowledge that the imposition of Alert levels may be used to judge the risk, that is fear, of infection and so encourage mask wearing in the methods section (lines 447 and 459)
Bearing in mind Alert levels were used as predictor variables in the regressions, the revisions to the theory section provide a stronger link between it and the results.
We believe the results do not add to theory (as such) but do provide empirical support for models (such as Bagozzi) that recognise the importance of implementation factors in the translation of intentions into action.
Reviewer 2 Report
Authors addressed several issued reported into my first review.
However, the Introduction does not effectively introduce the topic, as well as the Background is confused; to give an example, some references are over-described (e.g., 12) It is a question that concerns the whole world, and it is impossible to effectively describe its context by talking about a specific country, without also talking about pandemic and disease. I suggest write more detailed information, the cases reported for this study were relatively small compared to other countries. Furthermore, the COVID-19 pandemic has influenced all aspects of human life with an unprecedented global crisis characterized by drastic changes in social life, personal freedom, and economic activities and has created distress, as well as exacerbation of mental health issues in a traumatic stress context, especially for healthcare workers. Briefly, (in my opinion) the introduction needs to be extended, and the background better organized (many sub-sections bring confusion) and extended.
Section related to data and methodology is exhaustive.
“A Conclusion section is optional for this journal. The instructions to authors suggest it is only necessary if the Discussion section is unusually long and complex.” The conclusion is optional, however, if the authors include the conclusions, these must be written well… However, I only expressed my opinions.
Finally, authors responded other comments. Thanks for your response.
Major:
- Introduction, Background/Related works.
- The study is missing a comparison with other similar cases, or a description that reasons this absence.
Minors:
- The manuscript contains typos and grammar mistakes.
- References 64 and 65 report two time their numeration. (e.g., “64.64 aaa bbb”, “65.65 aaa bbb”).
Author Response
However, the Introduction does not effectively introduce the topic, as well as the Background is confused; to give an example, some references are over-described (e.g., 12) It is a question that concerns the whole world, and it is impossible to effectively describe its context by talking about a specific country, without also talking about pandemic and disease. I suggest write more detailed information, the cases reported for this study were relatively small compared to other countries. Furthermore, the COVID-19 pandemic has influenced all aspects of human life with an unprecedented global crisis characterized by drastic changes in social life, personal freedom, and economic activities and has created distress, as well as exacerbation of mental health issues in a traumatic stress context, especially for healthcare workers. Briefly, (in my opinion) the introduction needs to be extended, and the background better organized (many sub-sections bring confusion) and extended.
We have revised the Introduction to acknowledge the (for our generation) unprecedented nature of the Covid pandemic (lines 28-29). We feel there is no need to describe the crisis in detail as it has been widely covered already in this Journal and elsewhere.
We have noted in the Methods section (where we describe the case data) that less than 2000 cases of community transmission of Covid had been recorded prior to the second survey and most of these were in Auckland (lines 361-362).
We have explained reference [12] in some detail because (unlike other references on behavioural intentions) it includes a discussion of the factors that influence the translation of behavioural intentions into actual action, which is the central to this paper (see line 168 onwards).
- The study is missing a comparison with other similar cases, or a description that reasons this absence.
We are not aware of any study that compares behavioural intentions and actual behaviour in relation to mask wearing and COVID. Furthermore, we could not find a study that sought to explain differences in behaviour (implementation) in the context of similar intentions in any domain. Consequently, in the Discussion the only relevant literature we have been able to refer to is the literature on behavioural intentions in relation to COVID measures.
Minors:
- The manuscript contains typos and grammar mistakes.
- References 64 and 65 report two time their numeration. (e.g., “64.64 aaa bbb”, “65.65 aaa bbb”).
Typos, grammar, and reference numbering corrected.